# Caveolin-1-Mediated Tumor Suppression Is Linked to Reduced HIF1α S-Nitrosylation and Transcriptional Activity in Hypoxia

**DOI:** 10.3390/cancers12092349

**Published:** 2020-08-20

**Authors:** Carlos Sanhueza, Jimena Castillo Bennett, Manuel Valenzuela-Valderrama, Pamela Contreras, Lorena Lobos-González, América Campos, Sergio Wehinger, Álvaro Lladser, Rolf Kiessling, Lisette Leyton, Andrew F.G. Quest

**Affiliations:** 1Cellular Communication Laboratory, Center for studies on Exercise, Metabolism and Cancer (CEMC), Program of Cell and Molecular Biology, Institute of Biomedical Sciences (ICBM), Faculty of Medicine, Universidad de Chile, Santiago 8380453, Chile; csanhuezamunoz@yahoo.com (C.S.); jimenacastillob@gmail.com (J.C.B.); Orellana.paz@gmail.com (P.C.); america.camposg@gmail.com (A.C.); stevefoxster@gmail.com (S.W.); lleyton@med.uchile.cl (L.L.); 2Instituto Oncológico Fundación Arturo López Pérez, Santiago 7500921, Chile; 3Advanced Center for Chronic Diseases (ACCDiS), Santiago 8380000, Chile; manuel.valenzuela@ucentral.cl (M.V.-V.); lorotae@gmail.com (L.L.-G.); 4Laboratorio de Microbiología Celular, Instituto de Investigación e Innovación en Salud, Facultad de Ciencias de la Salud, Universidad Central de Chile, Santiago 8320000, Chile; 5Center for Regenerative Medicine, Faculty of Medicine, Clínica Alemana Universidad Del Desarrollo, Santiago 7710162, Chile; 6Thrombosis Research Center, Medical Technology School, Department of Clinical Biochemistry and Immunohaematology, Faculty of Health Sciences, Interdisciplinary Excellence Research Program on Healthy Aging (PIEI-ES), Universidad de Talca, Talca 3460000, Chile; 7Laboratory of Immunoncology, Fundación Ciencia & Vida; Facultad de Medicina y Ciencia, Universidad San Sebastián; Santiago 7780272, Chile; alladser@cienciavida.org; 8Immune and Gene Therapy Laboratory, Department of Oncology and Pathology, Karolinska Institutet, 17164 Stockholm, Sweden; rolf.kiessling@ki.se

**Keywords:** caveolin-1, hypoxia, HIF1α, S-nitrosylation, tumor suppression, VEGF

## Abstract

Caveolin-1 (CAV1) is a well-established nitric oxide synthase inhibitor, whose function as a tumor suppressor is favored by, but not entirely dependent on, the presence of E-cadherin. Tumors are frequently hypoxic and the activation of the hypoxia-inducible factor-1α (HIF1α) promotes tumor growth. HIF1α is regulated by several post-translational modifications, including S-nitrosylation. Here, we evaluate the mechanisms underlying tumor suppression by CAV1 in cancer cells lacking E-cadherin in hypoxia. Our main findings are that CAV1 reduced HIF activity and Vascular Endothelial Growth Factor expression *in vitro* and *in vivo*. This effect was neither due to reduced HIF1α protein stability or reduced nuclear translocation. Instead, HIF1α S-nitrosylation observed in hypoxia was diminished by the presence of CAV1, and nitric oxide synthase (NOS) inhibition by Nω-Nitro-L-arginine methyl ester hydrochloride (L-NAME) reduced HIF1α transcriptional activity in cells to the same extent as observed upon CAV1 expression. Additionally, arginase inhibition by (S)-(2-Boronoethyl)-L-cysteine (BEC) partially rescued cells from the CAV1-mediated suppression of HIF1α transcriptional activity. *In vivo*, CAV1-mediated tumor suppression was dependent on NOS activity. In summary, CAV1-dependent tumor suppression in the absence of E-cadherin is linked to reduced HIF1α transcriptional activity via diminished NOS-mediated HIF1α S-nitrosylation.

## 1. Introduction

Caveolin-1 (CAV1), a member of the caveolin family of proteins [1,2,3], is a scaffolding protein with a controversial role in cancer, given that it has been ascribed roles both as a tumor suppressor and a promoter of metastasis [4,5]. Work from this laboratory has shown that CAV1 sequesters β-catenin to the plasma membrane in a multi-protein complex with E-cadherin, thereby precluding the β-catenin/T-cell factor-Lymphoid enhancer factor (Tcf-Lef)-dependent transcription of genes, including survivin and cyclooxygenase-2 (COX-2), both of which are important for tumor cell survival, and in doing so, reducing cancer cell viability [6,7,8]. 

These functions of CAV1 *in vitro* are consistent with the notion that its activity as a tumor suppressor *in vivo* should require the expression of E-cadherin [9]. However, CAV1 suppresses tumor growth even in the absence of E-cadherin expression, albeit less efficiently [9]. 

Hypoxic areas (≤2% O_2_) [10] are widely present in solid tumors and adaptation to this hostile environment is frequently associated with enhanced metastasis and reduced patient survival [11,12,13,14]. Importantly, E-cadherin expression is severely reduced by hypoxia under low oxygen conditions [15,16]. Thus, hypoxia was considered an excellent model to evaluate CAV1 function in a manner independent of E-cadherin. 

Cells exposed to hypoxia activate the hypoxia-induced family (HIF) of transcription factors to permit metabolic adaptation, pH control and the neovascularization process, as well as epigenetic changes that facilitate cell adaptation to hypoxia [17,18]. Hypoxia-inducible factor-1α (HIF1α) is the most ubiquitous and best-described isoform of these transcription factors [19,20,21,22]. In normoxia, HIF1α is hydroxylated by prolyl hydroxylase enzymes (PHD) in key proline residues (P402 and P564), enabling recognition by the Von Hippel Lindau (VHL) ubiquitin–ligase complex, ubiquitination and subsequent proteasome-mediated HIF1α degradation [23,24]. In hypoxia, this process is inhibited, allowing for HIF1α accumulation and translocation to the cell nucleus, where it promotes the expression of several target genes, including vascular endothelial growth factor A (*VEGF*-A), glucose transporter 1 (*GLUT-1*) and lactate dehydrogenase (*LDH*) [25,26,27]. 

Interestingly, in this context, the loss of CAV1 expression in breast cancer tumor–stroma reportedly induces HIF1α activation [28] and the loss of CAV1 in stromal cells leads to a glycolytic and catabolic phenotype through HIF1α stabilization, favoring synergy with oxidative breast cancer cells [29]. Moreover, the analysis of tissue samples from CAV1 knockout mice revealed an increase in HIF1α target gene expression [30]. These observations suggest the existence of mechanism(s) by which CAV1 presence subdues HIF1α activation *in vivo*. 

Some of the best-characterized CAV1 target proteins to date are the nitric oxide synthases (NOS), which include the endothelial, neuronal and inducible isoforms, eNOS, nNOS, and iNOS, respectively [31,32,33,34,35]. In cancer, NO production is generally attributed to iNOS that is frequently overexpressed [36,37,38,39,40] and is associated with poor patient prognosis [41]. Importantly, elevated levels of NO have been shown to promote HIF1α stabilization, independent of oxygen levels [42,43]. Furthermore, NO released by macrophages causes HIF1α stabilization via the S-nitrosylation of cysteines present in the oxygen-dependent degradation domain of HIF1α [44]. In addition, HIF1α S-nitrosylation in the C-Terminal Transactivation Domain (C-TAD) of HIF1α promotes HIF transcriptional activity via enhanced HIF1α binding to the cAMP-responsive element binding protein (CREB) transactivation factor [44,45,46,47]. NO-derived metabolites can regulate HIF1α stabilization; for instance, S-nitrosoglutathione stabilizes, while peroxynitrite destabilizes HIF1α [48]. Taken together, these findings indicate that NO-mediated nitrosylation of HIF1α tends to stabilize the protein and favor HIF-dependent transcriptional activity. 

Thus, in this study, we evaluate whether E-cadherin-independent tumor suppression by CAV1 might be linked to the ability of CAV1 to reduce S-nitrosylation and, hence, HIF1α activation in hypoxia. 

## 2. Results

### 2.1. CAV1 Reduces HIF-Dependent Transcriptional Activity in Cancer Cells

To elucidate how CAV1 functions as a tumor suppressor in the absence of E-cadherin, we first evaluated E-cadherin protein levels in a panel of four different cancer cell lines. E-cadherin was neither detectable in HT29(US) or B16F10 cells stably transfected with pLacIOP (CAV1), nor in MDA-MB-231 cells transduced with an shRNA-targeting CAV1. Low levels of E-cadherin were detected in HEK293T cells, while DLD-1 protein extracts were included as positive controls for E-cadherin expression (Figure 1). Based on these observations, we chose to work with HT29(US) cells, a human colon cancer cell line with enhanced metastatic potential compared with commercially available HT29(ATCC) cells [7], highly metastatic B16F10 murine melanoma cells [9,49] (both with low CAV1 expression) and the human breast cancer cell line MDA-MB-231, which expresses high endogenous levels of CAV1 [50].

To evaluate the effect of CAV1 expression on HIF-dependent transcription, cancer cell lines were transfected with a HIF-gene reporter plasmid pGL3-HRE [51]. Thus, HT29(US) cells stably transfected with pLacIOP alone or the plasmid with a CAV1-encoding sequence (clones M1 and C14, respectively; [7,52]) were additionally transfected with the reporter plasmid pGL3-HRE and incubated for 24 h in hypoxia or normoxia (1% or 20% O_2_, respectively). In normoxia, basal levels of HIF-dependent transcription were low and no significant differences were observed when compared with HT29(US)(C14) or (M1) clones (Figure 2A). On the other hand, exposure to hypoxia increased ~12-fold for HIF-dependent transcription in HT29(US)M1 cells, while for HT29(US)C14, HIF transcriptional activity increased by only (roughly) five-fold (Figure 2A). Likewise, for HEK293T cells transiently transfected with pLacIOP(CAV1), HIF reporter activity decreased by approximately 50% in hypoxia (Appendix A). Taken together, these observations suggest that CAV1 expression reduces HIF transcriptional activity in hypoxia. 

To confirm our findings, we asked whether HIF-dependent transcription was modulated by the knockdown of CAV1 in cells expressing high endogenous levels of the protein. To this end, MDA-MB-231 breast cancer cells stably transfected with either the control (shLuc) or CAV1-specific (shCAV1) short hairpin RNA-expressing constructs were additionally transfected with the reporter plasmid pGL3-HRE and exposed to normoxia or hypoxia in order to evaluate changes in HIF transcriptional activity. As expected, in normoxia, CAV1 knockdown enhanced 3-fold basal HIF transcriptional activity (Figure 2B). Under hypoxia, reporter activity increased approximately 10-fold in shLuc cells, while for shCAV1 knockdown cells, a 50-fold increase in HIF-gene reporter activity was observed (Figure 2B). 

Taken together, these findings suggest that CAV1 reduces HIF transcriptional activity in hypoxia in human cancer cell lines of diverse origin. Additionally, we evaluated whether CAV1-mediated inhibition of the HIF-gene was associated with HIF1α target gene expression. Thus, we measured the mRNA levels of *VEGF-A*, *GLUT-1* and *LDH-A* by qPCR. In HT29(US) cells exposed to hypoxia (48 h), CAV1 expression was associated with a reduction in HIF1α-target gene expression. The most significant reduction observed was for *VEGF-A* and *LDH-A*, while *GLUT-1* was not significantly reduced (Figure 2C). Alternatively, CAV1 knockdown in MDA-MB-231 cells increased *VEGF-A* and *GLUT-1*, but not *LDH-A* expression in hypoxia (Figure 2D). Taken together, these results strongly indicate that CAV1 presence in different cancer cells reduces HIF1α activity and concomitantly target gene expression in hypoxia.

### 2.2. Reduced HIF1-Activation by CAV1 Is Not Due to Altered HIF1α Stability or Sequestration by CAV1

Next, we asked whether, in CAV1-expressing cells, HIF1α protein stability is affected in hypoxia. However, HIF1α protein levels induced by hypoxia were neither affected by CAV1 overexpression in HT29(US) colon cancer cells or by CAV1 knockdown in the breast cancer cell line MDA-MB-231 (Figure 3A,B). Similar data was obtained in HEK293T cells transiently transfected with the CAV1-encoding plasmid pLacIOP(CAV1) (Appendix A). HIF1α protein levels are controlled by proteasomal degradation [17]. Thus, one possibility was that HIF1α presence and turnover was altered in CAV1-expressing cells. To explore whether CAV1 expression may impact on the half-life of HIF1α, we evaluated protein levels in the presence of cycloheximide, added after chemical simulation of hypoxia using the iron chelator compound deferoxamine [53]. In initial experiments, we had attempted to estimate HIF1α half-life in cells expressing CAV1 or not following exposure to hypoxia; however, these efforts were unsuccessful because the half-life is very short, even in the absence of oxygen. The addition of deferoxamine leads to notable HIF1α expression levels and readily measurable protein turnover. Under these experimental conditions, HIF1α half-life was essentially identical (42 ± 27 vs. 46 ± 23 min) for HT29(US)M1 and C14 cells, respectively (Appendix A). Taken together, these results indicate that the CAV1-induced reduction in HIF1α activity in hypoxia is not likely attributable to either a reduction in HIF1α protein levels or an increase in HIF1α turnover.

The CAV1 protein contains a so-called scaffolding domain, implicated in the interaction of CAV1 with many proteins and their sequestration and/or functional inhibition [1,2]. Thus, we asked whether diminished HIF-gene reporter activity observed in CAV1-expressing cells may be the consequence of CAV1-mediated HIF1α protein sequestration. An analysis of cellular distribution by confocal microscopy showed no colocalization between CAV1 and HIF1α after either a 4 h or 24 h exposure to hypoxia (Figure 3C). Likewise, no differences were detected when HIF1α and CAV1 protein distribution were analyzed by subcellular fractionation (Figure 3D). Therefore, CAV1-mediated HIF1α inhibition is not the consequence of HIF1α sequestration or a reduction in HIF1α nuclear content, suggesting that CAV1-mediated HIF1α inhibition involved an indirect mechanism.

### 2.3. CAV1-Mediated HIF1α Inhibition Involves NOS Activity and Reduced HIF1α S-Nitrosylation in Hypoxia

The available evidence indicates that NO production and S-nitrosylation are relevant players in HIF1α stabilization and transcriptional activation [44,45,46,47]. To evaluate whether CAV1 expression is associated with these processes, HT29(US) colon cancer cells were treated with a general NOS inhibitor, Nω-Nitro-L-arginine methyl ester hydrochloride (L-NAME) (Figure 4A). In the presence of L-NAME, the hypoxia-induced increase in HIF1α transcriptional activity detected in cells lacking CAV1 (M1 cells) was reduced to the same extent as observed for the CAV1-expressing cells (C14 cells, *p* < 0.05; Figure 4A). Importantly, this effect was dependent on the concentration of L-NAME. Moreover, we inhibited arginase activity, since it represents an endogenous competitor of NOS enzymes for the use of L-Arginine [54,55]. We observed that the inhibition of arginase activity with 100 µM BEC (S-(2-boronoethyl)-l-cysteine) [56] partially rescued HIF1α-inhibition by CAV1 in HT29(US) cells in hypoxia (Figure 4B), suggesting that HIF1α transcriptional activity is modulated by an equilibrium between NOS and arginase activity under these conditions.

As mentioned previously, HIF1α levels/activities are modulated by S-nitrosylation [44,45,46,47]. Thus, we employed the biotin switch assay to assess HIF1α S-nitrosylation in HT29(US) cells. We found that the levels of HIF1α S-nitrosylation were reduced (~35%) in cells expressing CAV1 (C14) compared to Mock (M1) cells (Figure 4C). Taken together, these results suggest that CAV1-mediated HIF1α inhibition is likely due to CAV1-dependent modulation of HIF1α transactivation via reduced HIF1α S-nitrosylation under low oxygen conditions. 

### 2.4. CAV1 Tumor Suppression Function Involves NO In Vivo

We previously established an *in vivo* model using B16F10 melanoma cells in non-immunosuppressed syngeneic C57BL/6 mice to demonstrate that CAV1 functions as a tumor suppressor even in the absence of E-cadherin [9]. Thus, to corroborate our findings *in vivo*, we first needed to obtain evidence that CAV1 expression in B16F10 cells reduced HIF1α function in hypoxia. To that end, B16F10(Mock) and B16F10(CAV1) cells were exposed to hypoxia and HIF1α activation was quantified using the HIF gene-reporter assay. As anticipated based on the previous results, CAV1 expression significantly reduced HIF1α reporter activity in hypoxia, thereby validating the use of these cells for the following *in vivo* studies. Then, we injected these cells into mice and allowed the tumors to develop for 16 days before determining, by qPCR, the expression levels of *VEGF-A* in tumors formed by B16F10(Mock) or B16F10(CAV1) cells in syngeneic C57BL6 mice. In this case, increased *VEGF-A* expression was considered to be indicative of HIF activation *in situ* in the tumors. We observed reduced *VEGF-A* levels in B16F10(CAV1) tumors compared to those formed by B16F10(Mock) cells after 16 days of growth in C57BL/6 mice (Figure 5B), which we interpreted as being due to the reduced HIF1α-activation by CAV1 in hypoxia *in vivo*. As suspected, intraperitoneal L-NAME administration in C57BL/6 mice (7.5 mg/kg) reduced the growth of tumors formed by B16F10(Mock) cells to the same extent as observed for B16F10(CAV1) cells. Intriguingly, however, L-NAME administration to mice injected with B16F10(Mock) cells did not reduce tumor size beyond the effect attributable to CAV1 presence in B16F10(CAV1) cells (Figure 5C). This finding may be taken to indicate that inhibition of NOS in tumor cells, rather than systemic NOS inhibition, is what limits tumor growth by B16F10(Mock) cells.

We also evaluated the effect of CAV1 knockdown in B16F0 parental cells. B16F10 cells were initially obtained from B16F0 cells (spontaneous melanoma in C57BL/6 mice) by repetitive injection into the tail vein and recovery from the lungs of injected animals [57]. Like B16F10 cells, B16F0 cells also do not express E-cadherin. However, they do have detectable basal levels of CAV1 and display reduced metastatic potential compared to the B16F10 cells. In B16F0 cells, CAV1 expression was reduced ~80% after transduction with specific shRNA (Appendix A). Here, shRNA-mediated CAV1 knockdown enhanced tumor growth compared to B16F0(shLuc) control cells (Figure 5D). Interestingly, in the latter case, the treatment of mice with L-NAME prevented the increase in tumor growth observed for mice inoculated with B16F0(shCAV-1) cells (Figure 5D). Altogether, these findings indicate that CAV1 function as a tumor suppressor in the absence of E-cadherin, involves NOS inhibition, which reduces HIF1α S-nitrosylation and *VEGF* expression. Moreover, these results highlight the potential relevance of NOS activity in promoting tumor growth by favoring HIF-dependent *VEGF* expression in cancer cells. 

## 3. Discussion

CAV1 reportedly functions as a tumor suppressor in several models including colon [52], breast [58] and lung cancer [59], as well as melanoma [9] cells. Previous work from our group linked CAV1-mediated tumor suppression in the presence of E-cadherin to the inhibition of β-catenin/TCF-LEF-mediated transcription of survivin and COX-2 [6,7,8]. However, in the absence of E-cadherin, CAV1 still functions as a tumor suppressor, albeit less efficiently [9], indicating that additional mechanisms must be invoked to explain CAV1 function as a tumor suppressor. Xenograft experiments have shown that tumors are hypoxic and that although initial growth of cancer cells occurs to a limited extent in an avascular microenvironment [60,61], ultimately, the vascularization via HIF-dependent mechanisms is required to permit significant tumor growth. Thus, we evaluated how CAV1 expression in tumor cells lacking E-cadherin modulates responses in a hypoxic microenvironment where HIF is induced. We observed that CAV1 presence in cancer cells was associated with diminished HIF1α activity rather than protein levels and linked this to reduced nitrosylation in the presence of CAV1.

### 3.1. CAV1 Does Not Alter HIF1α Protein Levels in Hypoxic Cancer Cells

In hypoxia, after stabilization, HIF1α translocates to the nucleus to form a heterodimer with HIF1β, which binds to the Hypoxia Response Element (HRE) and promotes gene transcription [62]. Thus, potentially, CAV1 could have altered HIF1α stabilization and led to lower protein levels. However, our data show that CAV1 overexpression (HT29(US) and HEK293T) or knockdown (MDA-MB231 cells) neither altered HIF1α protein levels (Figure 3) in hypoxia nor the half-life of HIF1α after hypoxia simulation with deferoxamine (Appendix A). Thus, alternative mechanisms need to be invoked to explain CAV1-mediated HIF1α inhibition. 

### 3.2. CAV1 Inhibits HIF1α in Hypoxic Cancer Cells

Previous studies analyzing CAV1 knockout mice and stroma from breast cancer patients revealed that increased HIF1α target gene expression correlated with reduced CAV1 levels [28,30,63]. In gastric cancer cells, hypoxia (1% O_2_)-enhanced HIF1α stabilization coincides with the downregulation of CAV1 expression and epithelial–mesenchymal transition [64]. In agreement with these findings, we show here that CAV1 reduces HIF1α transcriptional activity and S-nitrosylation.

CAV1 is known to sequester a large number of proteins via interactions involving the Caveolin Scaffolding Domain [2]. Moreover, CAV1 is present in several subcellular locations including the nucleus of ovarian cancer cells [65,66]. However, we failed to detect any notable degree of colocalization between HIF1α and CAV1 following either short (4 h) or extended (24 h) exposure to hypoxia. Instead, both proteins were detected in different compartments (CAV1 in the cytoplasm; HIF1α in the nucleus) in hypoxia. Furthermore, the quantification of HIF1α protein distribution did not reveal significant differences in HIF1α nuclear content in CAV1-expressing cells (Figure 3). Thus, CAV1 expression does not reduce HIF1α nuclear translocation by favoring the retention of HIF1α in a cytoplasmic compartment, as has been reported for β-catenin [8]. These observations indicate that additional mechanisms are available by which CAV1 can modulate HIF1α in hypoxia—for instance, by altering HIF1α dimerization and/or HIF1α transactivation or HIF1α/DNA binding [67]. However, our results favor the notion that this should occur via a process that does not involve direct interaction between CAV1 and HIF1α, given that we did not detect any significant degree of colocalization between the two proteins.

### 3.3. CAV1 Modulates HIF1α-S-Nitrosylation in Hypoxia In Vitro

NO is important for HIF1α stabilization and transactivation activity in normoxia and hypoxia [42,43,44,68,69,70]. Moreover, low levels of NO are required for HIF1α stabilization in HCT116 colon cancer cells [68]. Also, CAV1 is a well-known inhibitor of all NOS isoforms, be it by direct interaction—for instance, with eNOS and nNOS [32,71]—or, as in the case of iNOS, by promoting proteasomal degradation [31,35]. Indeed, CAV1 overexpression reduces NO production in neuroblastoma SK-N-MC cells in hypoxia (2% O_2_) [72]. In our experiments, NOS inhibition with L-NAME reduced HIF1α gene reporter activity to a similar extent as CAV1 did, while treatment with BEC [54,73], partially rescued CAV1-mediated HIF1α inhibition in hypoxia, without affecting HIF1α gene reporter activity in HT29(US) cells lacking CAV1 (Figure 4). Furthermore, *in vivo* inhibition of NOS with L-NAME reduced tumor growth in cells lacking CAV1 to a similar extent as observed for CAV1-expressing cells. Alternatively, systemic treatment with L-NAME reduced tumor growth in mice inoculated with B16F0(shCAV1) cells (Figure 5).

Taken together, these results indicate that inhibition of NOS activity is likely to be responsible for reduced HIF1α activity and hence tumor suppression by CAV1. Interestingly, however, preliminary results obtained by mass spectrometry data indicated that arginase-1 co-immunoprecipitated with CAV1 in HT29(US) cells. Thus, an alternative scenario, consistent with the data obtained using the arginase inhibitor BEC, might be that CAV1 favors arginase activity, contributing thereby to reduced NO production and the nitrosylation of HIF1α. Additional studies are required to distinguish between these possibilities. 

S-nitrosylation has emerged as an important post-translational modification due to its capacity to modulate gene expression [47,74]. Conversely, HIF1α transactivation and stabilization are enhanced by S-nitrosylation. By using NO donors and NO-derived from NOS, Yasinska and Sumbayev [45] found that S-nitrosylation stimulates HIF1α transactivation. Moreover, in murine tumors exposed to ionizing radiation, increased NO production by macrophages leads to HIF1α S-nitrosylation [44]. Both groups showed that HIF1α is subject to the S-nitrosylation of C533 and C800, which are present in the Oxygen-Dependent Degradation (ODD) and C-TAD domains, respectively [44,45,46,47]. The S-nitrosylation of the ODD domain is thought to augment protein stability, while modification of the C-TAD domain favors transcription [44,45,46,47]. In our study, we showed that CAV1 reduces total HIF1α S-nitrosylation at, potentially, C533 and C800 in HT29(US) cells (Figure 4). However, considering that CAV1 did not alter HIF1α protein levels in hypoxia, it is intriguing to speculate that reduced HIF1α S-nitrosylation occurs preferentially in the C-TAD (Cys^800^) rather than the ODD (Cys^520^) domain of HIF1α in CAV1-expressing cells, thereby modulating HIF1α transcriptional activity rather HIF1α protein stability [45,75]. 

In summary (see model in Figure 6), our data show that CAV1 expression in colon and breast cancer cells reduces HIF1α transcriptional activity in hypoxia. This effect is linked to diminished HIF1α S-nitrosylation and, as a consequence, the transcription of *VEGF*, which likely limits tumor growth and contributes to the tumor suppressor function of CAV1 in the absence of E-cadherin. Importantly, our results also emphasize the relevance of NOS activity in promoting tumor growth by favoring HIF-dependent *VEGF* expression in cancer cells and the encouraging results obtained with L-NAME *in vivo* may provide an opportunity for the treatment of tumors that are highly dependent on angiogenesis.

## 4. Materials and Methods

### 4.1. Materials

The general NOS inhibitor Nω-Nitro-L-arginine methyl ester hydrochloride (L-NAME), iron chelator deferoxamine and the protein synthesis inhibitor cycloheximide were from Sigma (St. Louis, MO, USA). The arginase inhibitor BEC ((S)-(2-Boronoethyl)-L-cysteine, HCI) was from Calbiochem (Darmstadt, Germany). All reagents were dissolved in water, except for the *in vivo* experiments, where L-NAME was dissolved in sterile PBS. The primary antibodies anti-HIF1α (monoclonal) and anti-Lap2 (polyclonal) were purchased from BD Biosciences (San Jose, CA, USA). The anti-caveolin-1 (polyclonal) antibody was from Transduction Laboratories (Lexington, KY, USA). The polyclonal anti-β actin and anti-HSP90 antibodies were from Sigma (St. Louis, MO, USA). Secondary goat anti-rabbit and goat anti-mouse IgG coupled to horse raddish peroxidase (HRP) were from Bio-Rad (Hercules, CA, USA) and Sigma (St. Louis, MO, USA), respectively. Secondary antibodies conjugated to Cy3TM and fluorescein isothiocyanate (FITC) were from Jackson ImmunoResearch Laboratories (West Grove, PA, USA).

### 4.2. Cell Culture and Transfection

Cell lines stably transfected with the IPTG-inducible vector pLacIOP or pLacIOP(CAV1) [52], including HT29(US) colon adenocarcinoma cells (clone M1 and C14) and B16F10 murine melanoma (batch population Mock and CAV1) cells, were previously described [7,52]. High-endogenous CAV1-expressing MDA-MB231 breast cancer cells transduced with a shRNA against CAV1 (shCAV1) or scrambled-sequence (shRNA-scramble, shLuc) were described previously [50]. Moreover, B16F0 murine melanoma cells, expressing elevated basal CAV1 protein levels compared to B16F10 cells, were transduced with shRNA targeting CAV1 (shCAV1) or luciferase (shLuc) as a control. All cell lines were maintained as detailed previously [6,9,50]. Additionally, HT29(US) (1.8 × 10^5^) and MDA-MB-231 (7 × 10^4^) cells were transiently or stably transfected in suspension using Superfect (Qiagen, Germantown, MD, USA) following the manufacturer’s instructions. 

### 4.3. Plasmids

HIF-gene reporter plasmid pGL3-HRE, containing three copies of hypoxia-responsive-elements from the pyruvate dehydrogenase kinase (PDK) gene promoter, was generously donated by Dr. Kaye Williams (University of Manchester, UK) [51]. The pON249 plasmid overexpressing β-galactosidase [76], employed to normalize HIF-gene reporter assays, was provided by Dr. Sergio Lavandero (Universidad of Chile, Santiago, Chile). 

### 4.4. Hypoxia Treatment

Cells were exposed to hypoxia (1% O_2_, 5% CO_2_, 94% N_2_) [77] as previously described [78].

### 4.5. HIF-Gene Reporter Assay

HT29(US) and MDA-MB231 cells were transfected with 0.25 µg of pGL3-HRE and pON249 in 24-well plates. After treatment, cells were lysed, and luciferase and β-galactosidase activity were determined as described [6]. 

### 4.6. Western Blot

Total cell extracts were obtained as described [35] and nuclear protein extracts were obtained using the NE-PER Nuclear and Cytoplasmic Extraction Kit (Thermo Scientific, Rockford, IL, USA) following the manufacturer’s instructions. Protein extracts (70–100 µg) were separated by SDS-PAGE (8% gels) and transferred to nitrocellulose [79]. Blots were then probed with anti-HIF1α (1:150), anti-caveolin-1 (1:2500), anti-β actin (1:4000), anti-HSP90 (1:2000) or anti-Lap2 (1:2000) antibodies. Blots were subsequently incubated with secondary HRP-conjugated antibodies. Bound antibodies were detected with the EZ-ECL kit (Biological Industries, Kibbutz Beit Haemek, Israel) using the Discovery 12iC Ultralum System (Claremont, CA, USA). Protein levels were quantified by scanning densitometry. 

### 4.7. HIF1α Biotin Switch Assay

To detect S-nitrosylated HIF1α protein, the switch-assay kit (Cayman Chemicals, Ann Arbor, MI, USA) was employed according to the manufacturer’s instructions. Protein samples were then analyzed by SDS-PAGE and S-nitrosylated HIF1α was detected by immunoblotting with specific antibodies (see above).

### 4.8. Analysis of mRNA by PCR

Total RNA was extracted with TriZol (Invitrogen, Waltham, MA, USA) as previously described [8]. RNA was treated with DNAse (Promega, Madison, WI, USA), according to the manufacturer´s instructions, before qPCR. Samples were analyzed by quantitative PCR in a Stratagene Mx3000P thermocycler following the manufacturer´s instructions. cDNAs were amplified using the Maxima™ SYBR Green qPCR Master Mix (2X) (Fermentas, Vilnius, Lithuania). *VEGF-A*, *LDH-A*, *GLUT-1*, and β actin expression were calculated by the 2^−ΔΔCT^ method [80].

### 4.9. Indirect Immunofluorescence

After fixation, permeabilization and blocking [7], cells were subsequently incubated with anti-HIF1α (1:150) and anti-CAV1 (1:200) antibodies overnight at 4 °C. After washing, cells were incubated for 1 h with cyanin dye 3 (Cy3)-conjugated anti-mouse immunoglobulin G (IgG) (1:200) and FITC-conjugated anti-rabbit IgG (1:200). Additionally, the cell nucleus was stained with 4′,6-diamidino-2-phenylindole (DAPI) (0.4 μg/mL). Mowiol-mounted samples [6] were visualized in an Olympus Fluoview FV1000 confocal microscope (Shinjuku, Tokio, Japan). The objective and the numerical aperture used was UPLSAPO 60X, N.A 1.35, respectively. Images were merged and HIF1α staining was quantified using Image J software (NIH, Bethesda, MD, USA). HIF1α nuclear staining was quantified using the Region of Interest (ROI) manager tool (Image J). For quantification, seven or 14 random fields (normoxia and hypoxia, respectively), with at least four cells per field, were considered. All data were normalized to the area of the nucleus and expressed as the Fluorescence Intensity (pixels/μm^2^). 

### 4.10. Tumor Formation Assay

Animal care, handling and procedures were approved by the Bioethics committee (CBA#0417 FMUCH). C57BL/6 mice were inoculated in one flank with 3 × 10^5^ B16F10 cells (Mock and CAV1 cells) or B16F0 (shLuc and shCAV1) cells in 100 µL of physiological saline. Tumor size was measured using a caliper and tumor volume was calculated as previously described [9,49,52]. To evaluate the effect of NOS inhibition with L-NAME, C57BL/6 mice inoculated with B16F10 (Mock and CAV1) or B16F0 (shLuc and shCAV1) cells were systemically treated daily by intraperitoneal injections of L-NAME 7.5 mg/kg in a final volume of 100 µL. At the end of the experiments, mice were sacrificed following procedures approved by our institutional bioethics committee. 

### 4.11. Statistical Analysis

Data are expressed as the mean ± SEM of at least three independent experiments, unless otherwise indicated. The data was compared using a two-tailed unpaired t-test or two-way ANOVA with a Bonferroni post-test for multiple comparisons. A *p* value < 0.05 was considered a statistically significant difference.

## 5. Conclusions

CAV1 expression in cancer cells reduces HIF1α transcriptional activity in hypoxia by reducing HIF1α S-nitrosylation and, as a consequence, the transcription of *VEGF*, which likely limits tumor growth and contributes to the tumor suppressor function of CAV1 in the absence of E-cadherin. 

## Figures and Tables

**Figure 1 cancers-12-02349-f001:**
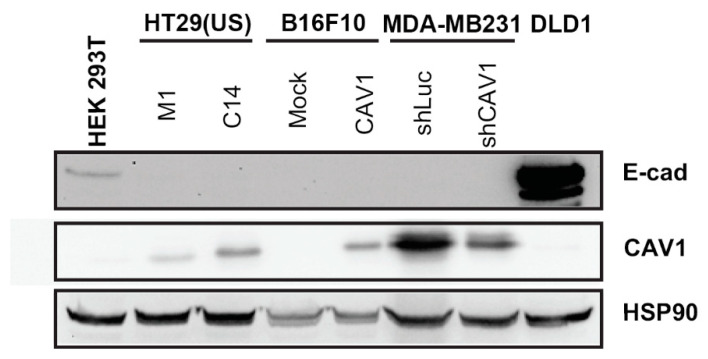
E-cadherin protein levels expressed in different cell lines. HEK293T, HT29(US) (clones M1 and C14), B16F10 (Mock and Caveolin-1 (CAV1)), MDA-MB231 (shLuc and shCAV1) and DLD-1 (positive control for E-cadherin expression) cell lines were maintained in culture at less than 70% confluency. To induce CAV1 expression, HT29(US) and B16F10 cells were incubated 24 and 48 h with isopropyl-b-D-thioglactopyranoside (IPTG) 1mM, respectively. Then, cells were lysed and cellular extracts were analyzed for the presence of E-cadherin (E-cad), CAV1, and HSP90 (loading control) by Western blotting.

**Figure 2 cancers-12-02349-f002:**
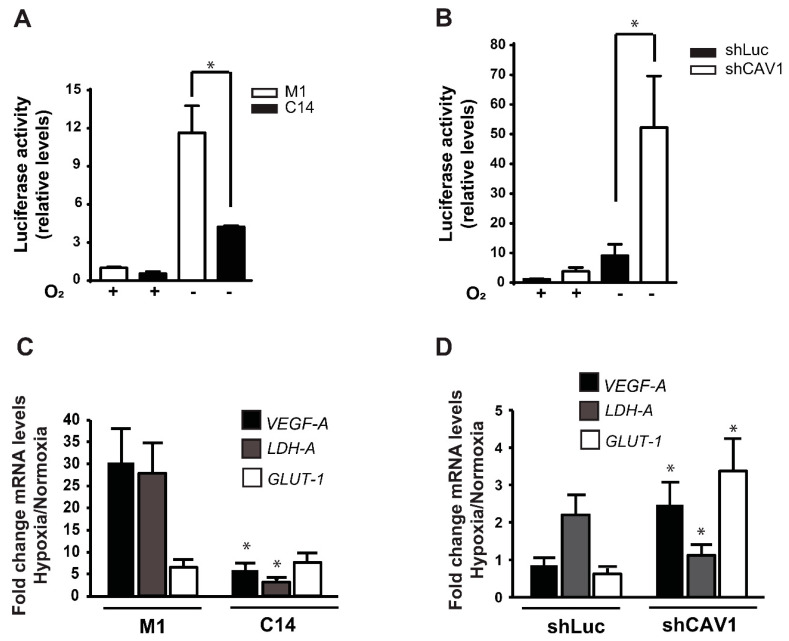
CAV1 reduces HIF-gene reporter activity and target gene expression in tumor cells. Hypoxia-induced family (HIF) gene-reporter assay in (**A**) HT29(US) M1 (white bars) and C14 (black bars) and (**B**) MDA-MB-231 (shLuc) (black bar) and (shCAV1) (white bar) cells in normoxia (+, 20% O_2_, 24 h) or exposed to hypoxia (−; 1% O_2_, 24 h). (**C**) Hypoxia-inducible factor-1α (HIF1α) target gene expression in HT29 (US) (M1) and (C14) in cells exposed to normoxia (20% O_2_, 24 h) or hypoxia (1% O_2_, 24 h). cDNAs from the indicated genes were amplified using primers listed in Table 1. *18S* was used as an internal control. The expression of HIF1α target genes (vascular endothelial growth factor A (*VEGF*-A), glucose transporter 1 (*GLUT-1*) and lactate dehydrogenase A (*LDH-A*)), was analyzed by qPCR in (**C**) HT29(US) and (**D**) MDA-MB-231 cells exposed to normoxia (20% O_2_, 24 h) or hypoxia (1% O_2,_ 24 h), as indicated. Bars represent fold changes in mRNA levels in hypoxia normalized to values obtained under normoxic conditions. Data are mean ± SEM from three independent experiments (* *p* < 0.05).

**Figure 3 cancers-12-02349-f003:**
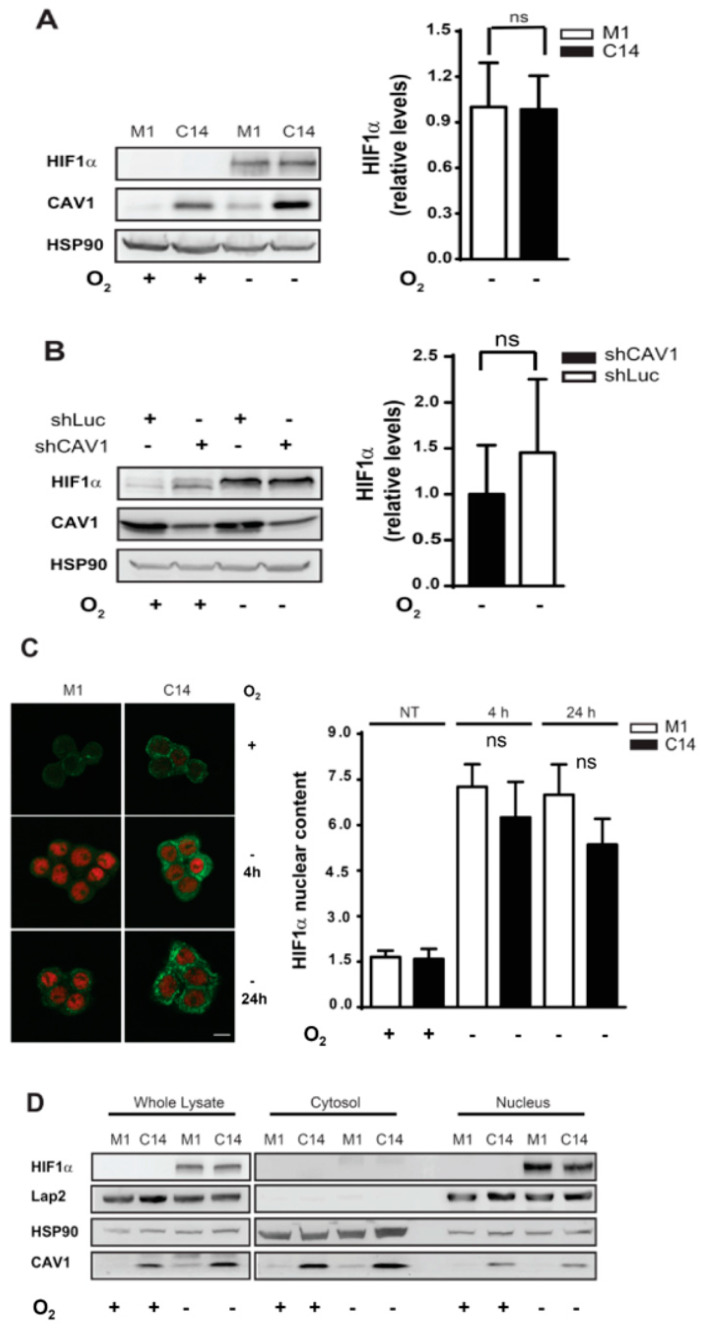
Suppression of HIF1α activity by CAV1 in cells exposed to hypoxia is independent of HIF1α degradation or sequestration. Cell extracts (50 µg total protein) were separated by sodium dodecyl sulfate polyacrylamide gel electrophoresis (SDS-PAGE) and then analyzed by Western blotting. HIF1α protein levels in A) HT29(US) (M1) (white bars) and (C14) (black bars) and B) MDA-MB-231 (shLuc) (black bars) and (shCAV1) (white bars) cells are shown in normoxia (+, 20% O_2,_ 24 h) or hypoxia (−; 1% O_2_, 24 h). HIF1α protein levels were quantified by scanning densitometry (**A**,**B**, graphs in panels to the right). (**C**) HIF1α subcellular localization in HT29(US) cells in normoxia (+, 20% O_2,_ 24 h) or hypoxia (−, 1% O_2_) for 4 h and 24 h. HIF1α (red) and CAV1 (green) were detected by indirect immunofluorescence analysis and confocal microscopy (magnification bar 10 µm). HIF1α nuclear staining for cells in normoxia (+, 20% O_2_, 24 h) or hypoxia (−; 1% O_2_ for 4 h and 24 h) was quantified using Image J software. HIF1α levels are shown in the graph (M1, white bars; C14 black bars); ns indicates that no significant differences were observed between M1 and C14 cells. (**D**) HIF1α subcellular distribution for HT29(US) (M1 and C14) cells in normoxia (+, 20% O_2,_ 24 h) or hypoxia (−, 1% O_2_, 24 h) was determined by the Western blotting of sub-cellular fractions obtained using the NE-PER Nuclear and Cytoplasmic Extraction Reagent (Thermo Scientific, Rockford, Il, USA). Lap2 and HSP90 protein levels were used as controls for nuclear and cytosolic fractions, respectively.

**Figure 4 cancers-12-02349-f004:**
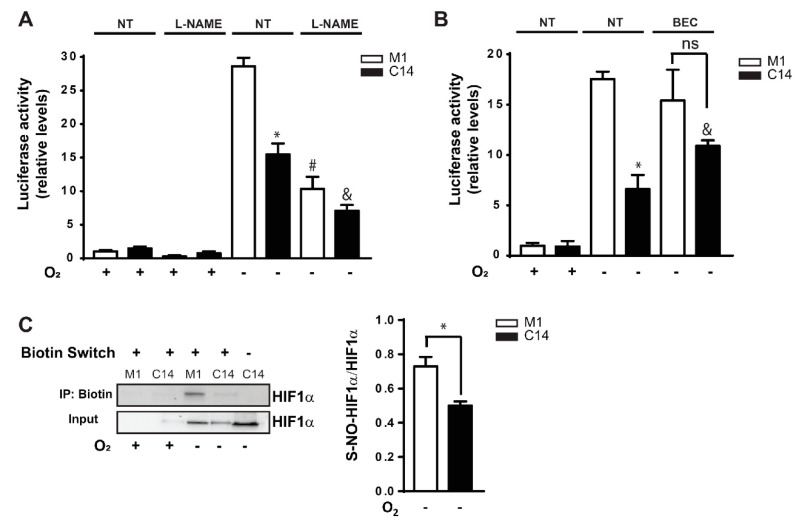
CAV1-mediated HIF1α inhibition involves alterations in nitric oxide synthase activity and HIF1α S-nitrosylation. (**A**) Effect of nitric oxide synthase (NOS) inhibition. HT29(US) cells (M1, white; C14, black bars) were incubated with the general NOS inhibitor L-NAME (10 mM) in normoxia (+, 20% O_2_, 24 h) or hypoxia (−, 1% O_2_, 24 h). HIF1α activation was quantified using the HIF gene-reporter assay (* *p* < 0.05, M1 hypoxia vs. C14 hypoxia; # *p* < 0.05, M1 hypoxia vs. M1 plus L-NAME hypoxia, & *p* < 0.05 C14 hypoxia vs. C14 plus L-NAME hypoxia). (**B**) Effect of the arginase inhibitor BEC. HT29(US) cells were treated with BEC (100 µM) in normoxia (+, 20% O_2_, 24 h) or hypoxia (−; 1% O_2_, 24 h). Then, HIF gene-reporter activity was assessed (* *p* < 0.05, M1 hypoxia vs. C14 hypoxia; no significant difference between M1 and C14 treated with BEC in hypoxia (ns). & *p* < 0.05, C14 hypoxia vs. C14 hypoxia plus BEC). (**C**) HIF1α S-nitrosylation. HT29(US) (M1 and C14) were incubated in normoxia or hypoxia, as indicated. Then, HIF1α S-nitrosylation was determined using the biotin switch assay. Western blots (left) representative of the results obtained in three independent experiments (quantification to the right) are shown (mean ± SEM; * *p* < 0.05).

**Figure 5 cancers-12-02349-f005:**
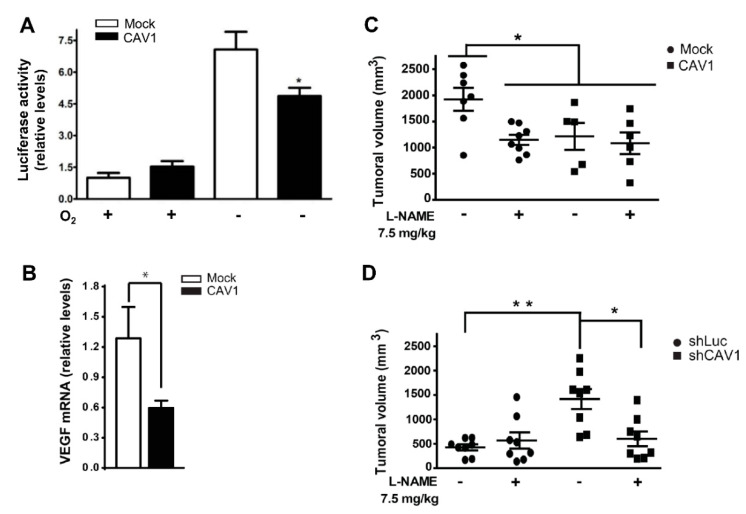
Effect of NOS inhibition on tumor growth. (**A**) HIF gene-reporter assay in B16F10(Mock) (white bars) or B16F10(CAV1) (black bars) cells in normoxia (+, 20% O_2_, 24 h) or exposed to hypoxia (−; 1% O_2_, 24 h). Data are the mean ± SEM from three independent experiments (* *p* < 0.05). (**B**) *VEGF* mRNA expression was determined by qPCR in B16F10(Mock) and (CAV1) tumors after 16 days of tumor growth in C57BL/6 mice. Values shown are from two separate experiments analyzing a total of eight B16F10(Mock) and 13 B16F10(CAV1) tumors (* *p* < 0.05). (**C**) The effect of systemic inhibition of NOS with L-NAME. C57BL/6 mice were injected subcutaneously with 3 × 10^5^ B16F10(Mock) or B16F10(CAV1) cells and then inoculated intraperitoneally (i.p) daily with the general NOS inhibitor L-NAME (7.5 mg/kg, final volume 100 µL) for 16 days. As a control, a group of mice were injected daily i.p with phosphate-buffered saline (PBS). (**D**) C57BL/6 mice were inoculated with 3 × 10^5^ B16F0(shLuc) or B16F0(shCAV1) cells. The mice were injected daily i.p with L-NAME (7.5 mg/kg, final volume 100 µL) for 18 days. In both cases (**C**,**D**) tumor volume (in mm^3^) was determined as indicated in Materials and Methods. Data are averages (mean ± SEM) of results obtained with *n* = 8 mice per group (* *p* < 0.05; ** *p* < 0.01).

**Figure 6 cancers-12-02349-f006:**
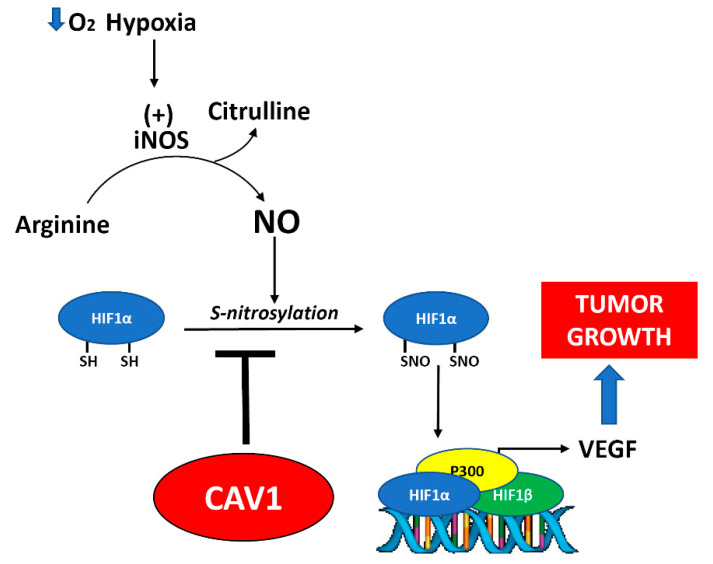
Working model summarizing the main findings in this study. CAV1 acts as a tumor suppressor in the absence of E-cadherin. The effect of CAV1 reported here is not mediated by reducing protein stability or nuclear translocation, but rather by the S-nitrosylation of HIF1α. Based on our main findings, we propose that HIF1α S-nitrosylation induced by hypoxia is inhibited by CAV1, preventing VEGF induction, thereby providing a rationale for E-cadherin-independent tumor suppression by CAV1.

**Table 1 cancers-12-02349-t001:** Primers used for RT-PCR and qPCR analysis.

Gene	Primer	Sequence 5′ → 3′
*VEGF-A*	SN	ATCCGGGTTTTATCCCTCTTC
	AS	TCTCTCTGGAGCTCTTGCTA
*LDH-1*	SN	ACGTCAGCAAGAGGGAGAA
	AS	TCTTCCAAGCCACGTAGGT
*GLUT-1*	SN	AAGGAAGAGAGTCGGCAGAT
	AS	TCGAAGATGCTCGTGGAGTA
*β Actin*	SN	AAATCGTGCGTGACATTAAGC
	AS	CCGATCCACACGGAGTACTT

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
