# Peer review of "Caveolin-1-Mediated Tumor Suppression Is Linked to Reduced HIF1α S-Nitrosylation and Transcriptional Activity in Hypoxia"

_cancers, 2020, doi:10.3390/cancers12092349_

Round 1

Reviewer 1 Report

The manuscript is based on a defined premise of evaluating CAV1’s role in tumor suppression in the absence of E-Cadherin. The concept is supported by the notion that E-Cadherin expression is low in hypoxia as stated by the authors in the introduction. The introduction and methods were well described. Results section indicates some lacunae which weakens the approach undertaken in few sections of results. Especially, switching cell line models between in vitro and in vivo experiments and lack of evaluations in tumor studies are some weaknesses to be rectified to make the study more efficient. Justification of observed results and approach will clarify these issues.

Abstract:

Abstract does not reflect the suppression of HIF1 in hypoxic conditions observed in present study. Please modify the abstract statements to indicate the hypoxic conditions were used in these studies.

Results:

  • Please use italics in figure 2 for gene expression.
  • Lines 137-138: Cell lines of diverse origin is misleading as the experiments were performed in the context of cancer cells. Please correct it!
  • Using different cell line models for in vitro vs in vivo experiments does not substantiate the models. Same cell lines should be included to avoid controversies.
  • HIF1 evaluations from tumor studies are missing. Assumptions may not support the hypothesis!
  • Results presented in Fig.4A do not support the in vivo results observed in Fig. 5B. When NAME was used M1 and C14 showed reduced HIF1 activities. The result is not the same in Fig. 5B. Please justify!
  • Quantification of NO from in vitro studies would add more support to the results section.
  • Please include the cell line model used in each panel of figure legends.

Discussion

  • Though, E-cadherin null models were used in the present study; mention about E-cadherin and NOS in tumor models should be made in the discussion.
  • Results in Fig.3 indicate lack of colocalization of HIF1aand CAV1. These results have to be compared and discussed based on previous reports indicating such colocalization especially in hypoxic conditions.
  • Line 298: 3. CAV1 modulates HIF1α-S-nitrosylation in hypoxia. This header and following description of in vivo results is misleading as the header only suits for in vitro studies. Further evaluations of S-nitrosylation in tumors is missing to reach the conclusions drawn from in vivostudies. 
  • Line 312: Please modify HIFa

Author Response

Reviewer#1

Comment: Abstract does not reflect the suppression of HIF1 in hypoxic conditions observed in present study. Please modify the abstract statements to indicate the hypoxic conditions were used in these studies.

Response: This has been correted and hypoxia is now mentioned in the abstract.

Comment: Please use italics in figure 2 for gene expression.

Response: This has been corrected as requested.

Comment: Lines 137-138: Cell lines of diverse origin is misleading as the experiments were performed in the context of cancer cells.

Response: This has been corrected.

Comment: Using different cell line models for in vitro vs in vivo experiments does not substantiate the models. Same cell lines should be included to avoid controversies.

Response: Indeed, the reviewer is correct. There was a lack of connection between the in vitro and in vivo studies because no data was included using B16F10 cells in vitro. This has now been remedied by showing in B16F10 cells that CAV1 expression reduces HIF reporter activity when the cells were subjected to hypoxia (Figure 5 A). In addition, we provided an introduction to this section justifying the switch in cell models.

Comment: HIF1 evaluations from tumor studies are missing. Assumptions may not support the hypothesis!

Response: In Figure 5 B, we provide evidence that in tumors VEGF expression is reduced in samples isolated from tumors formed by CAV1 expressing B16F10 cells as compared to the B16F10(Mock) cells. In the previous Figures, we established in vitro that CAV1 reduced HIF activity and VEGF expression to a similar extent in hypoxic cells. Moreover, the extent of VEGF expression is considered an appropriate indicator of hypoxia in the tumor environment. Thus, we feel evaluating VEGF expression represents an good readout for HIF induction and activity in tumors.

Comment: Results presented in Fig.4A do not support the in vivo results observed in Fig. 5B (now 5C). When NAME was used M1 and C14 showed reduced HIF1 activities. The result is not the same in Fig. 5B. Please justify!.

Response: The reviewer is correct in that we saw a difference in HIF reporter activity between HT29(US) cells with CAV1 and these same cells treated with L-NAME (10 mM) (see Fig. 4 A). The effect of L-NAME was very much concentration dependent. So for instance L-NAME (3 mM) was already significantly less effective in reducing HIF reporter activity. Also, at this concentration L-NAME did not have an effect beyond that seen upon expression of CAV1 (data not shown). In the tumor formation assays using B16F10 cells, no additional effect of L-NAME (7.5 mg/kg) beyond that seen upon expression of CAV1 was detected (now Fig. 5 C). When defining the dose to be used in vivo, we saw similar reductions in tumor growth with 7.5 and 15 mg/kg. With 30 mg/kg the effects we slightly more pronounced, however there was greater variability, suggesting we might be seeing off-target effects at higher concentrations. Thus, we opted for lowest dose 7.5 mg/kg in the in vivo experiments shown. The problem is that we have no way to compare directly in vitro (10 mM) and in vivo (7.5 mg/kg) results. It is possible that the 7.5 mg/kg in vivo represent a borderline concentration with an effect similar to that of 3 mM in vitro in which case we would not expect a difference.

Comment: Quantification of NO from in vitro studies would add more support to the results section.

Response: We tried to set up the measurement of NO in vitro, but unfortunately the student working on the project left the lab after finishing his thesis and the person who followed was unable to get meaningful results.

Comment: Please include the cell line model used in each panel of figure legends

Response: We have checked to ensure that the cell lines used in each case are indicated.

Comment: Though, E-cadherin null models were used in the present study; mention about E-cadherin and NOS in tumor models should be made in the discussion.

Response: Sorry, it is not clear to us how we should integrate such information in the discussion, given that we specifically chose cancer cell lines  lacking E-cadherin. However, we also did some experiments with the immortilizaed HEK293T human embryonic kidney cells (see Supplementary Figure 1) which express low but detectable levels of E-cadherin (see Figure 1) and got similar results in that CAV1 expression also reduced HIF reporter activity in these cells exposed to hypoxia. So, this can be taken to indicate that the presence of limited amounts of E-cadherin do not impede the ability of CAV1 to reduce HIF activity in hypoxia.

Comment: Results in Fig.3 indicate lack of colocalization of HIF1aand CAV1. These results have to be compared and discussed based on previous reports indicating such colocalization especially in hypoxic conditions.

Response: Unfortunately, the reviewer did not indicate specifically which references in the literature he/she is refering to and a search looking for “caveolin-1 and HIF1a colocalization” did not yield any relevant results. Thus, unfortunately, we were not able to add anything along these lines to the discussion.

Comment: Line 298: 3. CAV1 modulates HIF1α-S-nitrosylation in hypoxia. This header and following description of in vivo results is misleading as the header only suits for in vitro studies. Further evaluations of S-nitrosylation in tumors is missing to reach the conclusions drawn from in vivostudies. 

Response: We agree, this was an overstatement. We have now modified the title in 3.3 such as to refer exclusively to the nitrosylation in vitro.

Comment: Line 312: Please modify HIFa.

Response: We have checked to make sure that HIF1a appears as HIF1alpha.

Reviewer 2 Report

Congratulations to the authors. This is an interesting study and the results are very interesting. This study aimed to evaluated whether E-cadherin-independent tumor suppression by CAV1 might be linked to the ability of CAV1 to reduce S-nitrosylation and hence, HIF1α activation in hypoxia. Also, the authors deal with an interesting and important topic from a scientific point of view. the experiments in this study are well constructed and presented. The authors use CAV1 expression in colon and breast cancer to reduce HIF1α S-nitrosylation and, as a consequence, transcription of VEGF to limits tumor growth. Nevertheless, there are several minor comments that should be addressed:

INTRODUCTION

-Line 52: “low oxygen conditions”. Hypoxia per se is a reduction in the oxygen tension of organs, tissues, or cells is caused by an imbalance between oxygen supply and consumption. What level of hypoxia (%) do the authors with low refer to? Please specify it.

RESULTS

-Figures in general: 1% O2 is always set on the x-axis of the figures, however this can be confusing because the - sign refers to normoxia (20% O2). I would recommend the authors to set only (% O2) on the X axis.

-Figure 2: In figures C and D I don't see the - and + sign, corresponding to normoxia and hypoxia, respectively. What condition do the values refer to?

-Figure 3A and 4C: In the bars figure it is necessary to put the symbol of the oxygenation condition.

-Figure 4B: If there is no significance between the two conditions, it would be better not to put (NS). It could make it difficult to read this figure. Please apply it in all similar situations.

Supplementary material

Supplementary Figure 1: I think it is necessary to specify section a) in the explanation.

Supplementary Figure 2: Delete “a” at the end of GraphPad Prism.

DISCUSSION

The discussion is well written and provides a lot of information about the present study and similar ones.

-Line 321: Please, reference number should be placed after Yasinska and Sumbayev.

METHODS

The methods section is very clear and consistent. Congratulations to the authors.

Author Response

Reviewer #2

Comment: Congratulations to the authors. This is an interesting study and the results are very interesting…..

Response: we would like to thank the reviewer for the positive appreciation of our study.

Comment: Line 52: “low oxygen conditions”. Hypoxia per se is a reduction in the oxygen tension of organs, tissues, or cells is caused by an imbalance between oxygen supply and consumption. What level of hypoxia (%) do the authors with low refer to? Please specify it.

Response: This has now been defined in the introduction and a reference was added.

Comment: Figures in general: 1% O2 is always set on the x-axis of the figures, however this can be confusing because the - sign refers to normoxia (20% O2). I would recommend the authors to set only (% O2) on the X axis.

Response: We have corrected our figures as suggested. Indeed, in retrospect the nomenclature we were using was very confucing. Now we refer to 20 % O2 as normoxia (+) and 1% O2 as hypoxia (-).

Comment: Figure 2: In figures C and D I don't see the - and + sign, corresponding to normoxia and hypoxia, respectively. What condition do the values refer to?

Response: Please note that in Figure 2 C and D we are showing normalized data, this is to say values obtained in hypoxia divided by those registered in normoxia.

Comment: Figure 3A and 4C: In the bars figure it is necessary to put the symbol of the oxygenation condition.

Response: this has now been corrected as requested.

Comment: Figure 4B: If there is no significance between the two conditions, it would be better not to put (NS). It could make it difficult to read this figure. Please apply it in all similar situations.

Response: We feel it is important to indicate whether variations between values are significant or not. Particularly in the case of Figure 4 B, this is relevant because the symbol & may lead readers to think that the apparent difference M1 to C14 is significant.

Comment: Supplementary Figure 1: I think it is necessary to specify section a) in the explanation

Response: This has been corrected.

Comment: Supplementary Figure 2: Delete “a” at the end of GraphPad Prism.

Response: This has been corrected.

Comment: The discussion is well written and provides a lot of information about the present study and similar ones.

Response: thanks for the positive feedback.

Comment: Line 321: Please, reference number should be placed after Yasinska and Sumbayev.

Response: this reference has now been cited correctly.

Comment: The methods section is very clear and consistent. Congratulations to the authors.

Response: Again thanks for the positive feedback.

Reviewer 3 Report

In this paper by Sanhueza et al., authors have evaluated the mechanisms underlying tumor suppression by Caveolin-1 (CAV1) in cancer cells lacking E-cadherin. They found that CAV1 reduced HIF activity and Vascular Endothelial Growth Factor expression in vitro and in vivo, via diminished NOS-mediated HIF1α S-nitrosylation.

Authors made many affords to demonstrate the mechanisms of action involved and the results obtained by effective and convincing methods seem consistent with the hypothesis of the rationale.

My only doubt is about the use of HSP90 as a loading control in western blot experiments reported in Figs. 1 and 3. Authors should explain better the use of this protein, even because it seems to be at least slightly affected by treatments in Fig.3.

Author Response

Reviewer #3

Comment: Authors made many affords to demonstrate the mechanisms of action involved and the results obtained by effective and convincing methods seem consistent with the hypothesis of the rationale.

Response: we thank the reviewer for the positive assessment of our manuscript.

Comment: My only doubt is about the use of HSP90 as a loading control in western blot experiments reported in Figs. 1 and 3. Authors should explain better the use of this protein, even because it seems to be at least slightly affected by treatments in Fig.3. 

Response: At some stage during Carlos’ thesis, we were having problems with the antibody against b-actin (we had to change the vendor) and so Carlos switched to HSP90, which people in other laboratories in the center were using successfully as a loading control. In general, this worked well and no significant differences in the expression of this protein between normoxic and hypoxic conditions were detected.